# Characterization of Phytohormones and Transcriptomic Profiling of the Female and Male Inflorescence Development in Manchurian Walnut (*Juglans mandshurica* Maxim.)

**DOI:** 10.3390/ijms23105433

**Published:** 2022-05-13

**Authors:** Xiang Li, Rui Han, Kewei Cai, Ruixue Guo, Xiaona Pei, Xiyang Zhao

**Affiliations:** 1College of Forestry and Grassland, Jilin Agricultural University, Changchun 130118, China; lx2016bjfu@163.com (X.L.); hanrui4579@jlau.edu.cn (R.H.); 2State Key Laboratory of Tree Genetics and Breeding, School of Forestry, Northeast Forestry University, Harbin 150040, China; ckwnefu@163.com; 3College of Horticulture, Jilin Agricultural University, Changchun 130118, China; ruixueguocc@163.com

**Keywords:** *Juglans mandshurica*, flower, transcriptome analysis, phytohormone, transcription factor

## Abstract

Flowers are imperative reproductive organs and play a key role in the propagation of offspring, along with the generation of several metabolic products in flowering plants. In *Juglans mandshurica*, the number and development of flowers directly affect the fruit yield and subsequently its commercial value. However, owing to the lack of genetic information, there are few studies on the reproductive biology of *Juglans mandshurica*, and the molecular regulatory mechanisms underlying the development of female and male inflorescence remain unclear. In this study, phytohormones and transcriptomic sequencing analyses at the three stages of female and male inflorescence growth were performed to understand the regulatory functions underlying flower development. Gibberellin is the most dominant phytohormone that regulates flower development. In total, 14,579 and 7188 differentially expressed genes were identified after analyzing the development of male and female flowers, respectively, wherein, 3241 were commonly expressed. Enrichment analysis for significantly enriched pathways suggested the roles of MAPK signaling, phytohormone signal transduction, and sugar metabolism. Genes involved in floral organ transition and flowering were obtained and analyzed; these mainly belonged to the M-type MADS-box gene family. Three flowering-related genes (*SOC1*/*AGL20*, *ANT*, and *SVP*) strongly interacted with transcription factors in the co-expression network. Two key *CO* genes (*CO3* and *CO1*) were identified in the photoperiod pathway. We also identified two *GA20xs* genes, one *SVP* gene, and five *AGL* genes (*AGL8*, *AGL9*, *AGL15*, *AGL19*, and *AGL42*) that contributed to flower development. The findings are expected to provide a genetic basis for the studies on the regulatory networks and reproductive biology in inflorescence development for *J. mandshurica*.

## 1. Introduction

The flower is an important organ contributing to the growth and development of flowering plants and is closely related to the fruiting character, plant yield, and quality [1,2]. Therefore, understanding the molecular mechanisms that induce flowering is crucial for plant breeding, especially for the species with special mating types, including heterodichogamy. Flowering is an imperative sign for the transition from vegetative to reproductive growth phases, which ordinarily includes multiple biological processes of flowers such as bud differentiation, organ development, and morphogenesis [3]. Herein, the flowering phenology of male and female flowers at a specific time ensures normal pollination and seed/fruit development. At present, studies on flowering are research hotspot for many model species and crucial horticultural plants [4,5,6]. In particular, for the model plant, *Arabidopsis thaliana*, several regulators involved in the flowering induction have been identified through studies on the ambient temperature, vernalization, photoperiod, gibberellin (GA), autonomous, age-dependent, circadian clock, and trehalose-6-phosphate (T6P) pathway [7]. The regulators involved in the aforementioned pathways can integrate signaling transduction and gene interactions, along with the floral integrator genes to facilitate the transition from vegetative to the reproductive meristem, thereby activating the expression of flowering genes including *FLOWERING LOCUS C* (*FLC*), *FLOWERING LOCUS T* (*FT*), *FRUITFULL* (*FUL*), *SUPPRESSSOR OF OVEREXPRESSION OF CONSTANS1* (*SOC1*), *APETALA1* (*AP1*), *CONSTANS* (*CO*), *LEAFY* (*LFY*), and *TARGET OF EAT1* (*TOE1*) [8,9]. Among them, *LFY* is activated by the photoperiod and GA pathways, and it not only regulates the flowering time but also induces the formation of the floral meristem [10]. The regulation of *FT* gene expression in the photoperiod pathway is influenced by *CO*, which further activates *SOC1*, thereby promoting flowering [11,12]. *FT* and *LFY* can synergistically induce the expression of *AP1*, thereby improving flowering in *A. thaliana* [13].

Flowering is a complex process that integrates multiple regulatory networks. Previous studies show that environmental (including photoperiod, temperature, and nutritional status) and hereditary factors (including phytohormones, gene regulation, and developmental stage) mediate the transition to flowering in plants [14]. During the process of evolution, plants have gradually evolved sensitivity to different temperatures and light for better adaptation to their environment, thereby mediating transition to flowering [15,16]. In *Arabidopsis*, a decrease in the ambient temperature (to less than 16 °C) causes delayed flowering activity, while vernalization signaling significantly enhances flowering control [17]. Additionally, flowering is induced by changes in the ambient temperature in species of *Phalaenopsis*, and low ambient temperature (less than 26 °C) promotes the flowering process, while the activity can be reversed at elevated ambient temperature [18]. Photoperiod is a key factor for flowering activity and its specific function in *Arabidopsis* and rice is known and has been initially studied in other non-model species [19,20,21]. For instance, the *Zea CENTRORADIALIS 8* (*ZCN8*) gene encoding *FT* florigen is a good candidate for integrating photoperiods and endogenous to induce flowering in maize [22]. Furthermore, the effects of many phytohormones such as GA, abscisic acid (ABA), and auxin have been confirmed. Hormone signaling transduction is involved in the flowering processes in *Arabidopsis* [23,24]. Accumulating evidence for the GA pathway in controlling the flowering activity suggests that there is a significant difference in GA regulating flowering in different plants [25,26,27]. Moreover, different GA hormone types (including GA_3_, GA_4_, GA_5_, GA_6_, and GA_7_) specifically activate discrepant flowering induction processes in the same species [28]. For example, GA_4_ positively regulates flowering in *Malus domestica*, whereas GA_3_ exerts reversion effects [29]. The application of GA promotes the expression of *SOC1* and *LFY*, which in turn improves the expression of downstream flowering genes, thereby ensuring normal flower morphogenesis [30]. Notably, although the mechanisms underlying flowering in model (often annual) species have been widely studied, research on perennial woody plants remains lacking.

Manchurian walnut (*Juglans mandshurica* Maxim.) is a perennial woody species, with economic and ecological value. The distribution of *J. mandshurica* is in Japan, North Korea, the Russian Far East, and northeast China [31,32]. In particular, it possesses the characteristics of multipurpose components, long life, and heterodichogamy. Under natural conditions, flower of *J. mandshurica* is monoecious and exhibits a high degree of morpho-differentiation in color, size, and morphology [33,34]. Male flowers are in catkins of sizes approximately between 9 and 20 cm, while four to ten female flowers form a spike. Flowering in *J. mandshurica* can in general be grouped into two clear mating types, protogynous and protandrous, and these are randomly distributed in specific populations. The synchronization and non-interference of mating types not only effectively avoid selfing but also improve the outcrossing rates in natural pollination of *J. mandshurica*. At present, only a few studies have investigated the reproductive biology of *J. mandshurica* in relation to its influences on flower development [33]. Understanding the genetic mechanisms and regulatory networks underlying the transition to flowering would contribute to the innovations in crossbreeding and polyploid breeding of *J. mandshurica*.

In this study, the flower developmental processes and coordination mechanisms of *J. mandshurica* were analyzed by RNA-sequencing. Using enzyme immunoassay, the phytohormones in three developmental stages of male and female inflorescences were identified and assessed. Furthermore, we constructed a gene transcriptional regulatory network based on the transcriptomic profiles. Additionally, we investigated the expression patterns of key regulatory genes involved in phytohormone signal transduction, MAPK signaling, starch and sucrose metabolism, and glycolysis/gluconeogenesis pathways during the developmental processes of female/male inflorescences. The results are expected to provide new insights into regulatory mechanisms underlying the development of female/male flowers and lay a fundamental reference for further protection and utilization of this precious plant species.

## 2. Results

### 2.1. Morphological Characteristics of Female/Male Flowering Transitions

*J. mandshurica* is a monoecious plant with unisexual flowers (Figure 1a,b). According to the unique morphological changes during the transition to flowering, the continuous male flower development was divided into three key stages including the dormant bud (MS1), anther formation (MS2), and anther maturation (MS3), while the continuous female flower development was divided into dormant bud (FS1), female bud formation (FS2), and flowering stages (FS3) (Figure 1c).

In the dormancy of bud in winter, the male/female bud showed rust yellow coloration. In particular, the male inflorescence bud of *J. mandshurica* was conical and measured less than 1 cm in length, while the female inflorescence bud was irregular with a specific bract, and measured greater than 1 cm. The axis of catkin gradually elongated and the anther formation began from an immature stage. During the anther formation, the color of the male inflorescence (rust yellow to green) and floral axis length displayed obvious changes, and thus, this stage was also regarded as the rapid development stage. Subsequently, in the late developmental stage of the male inflorescence, the length of its axis was found to be longer than 15 cm and the mature anthers were formed. In the female inflorescence, when the inflorescence bud began to develop, a growth point was observed, which indicated that the female inflorescence had transformed from a dormant bud to a flowering bud. By flower bud breaking and bract elongation, the organs of the stigma began to crack and approximately 4–10 female flowers emerged in the inflorescence axis. The stigmata of the female inflorescence showed distinctly visible magenta coloration. The above results suggested that the morphological characteristics of the male/female inflorescence changed substantially during the transition to flowering (Figure 1c).

### 2.2. Changes in Phytohormones during the Flowering Transition Process

To investigate the differences in phytohormones during the transition to flowering, content analyses were performed at three pivotal stages for male and female inflorescence separately. In the *J. mandshurica* female flowers, the abscisic acid (ABA) content differed significantly (*p* < 0.05) among the three developmental stages and decreased progressively from FS1 to FS3. The gibberellins (GA) content decreased significantly (*p* < 0.05) between FS1 and FS2 stages but almost remained unchanged from FS2 to FS3. In addition, the content of auxin (IAA) and jasmonic acid (JA) showed relatively high levels in the early developmental stages. In particular, there were no significant differences in the levels of brassinosteroids (BR), cytokinin (CTK), ethylene (ETH), and zeatin (ZT) from FS1 to FS3 stages (Figure 2a). During the development of the male inflorescence, the results suggested that the content of ABA and BR were maintained at high levels through the MF3 stage, and there was a significant difference (*p* < 0.05) among the different stages of transition to flowering. In particular, the GA content in MS1 was consistent with that in MS2 but increased rapidly from MS2 to MS3. However, ETH, IAA, and ZT showed high level at the MS1 stage. The CTK and JA levels showed no substantial differences among the stages (Figure 2b). 

### 2.3. Identification and Functional Enrichment of DEGs during the Development of Male Inflorescence

To assess the transcriptional profiles at different developmental stages of the male flower, a total of nine cDNA libraries with three biological replicates for each stage were constructed and used for RNA-seq. In total, 490,971,372 raw reads were obtained. Next, after filtering and sequencing error checks for raw data, a total of 468,319,748 clean reads were used for subsequent bioinformatic analyses. Cumulatively, the average Q20, Q30, and GC content of the clean reads were 97.05%, 92.16%, and 45.86%, respectively, suggesting a high quality of sequencing (Appendix A). Furthermore, an average of 96.00% clean reads were mapped at the chromosomal level of the *J. mandshurica* genome, suggestive of a high-level mapping rate (Appendix A). The result of the principal component analysis (PCA) for all arrays using the FPKM value is shown in Figure 3a. The biological triplicates for each developmental stage showed excellent grouping, and three male inflorescence developmental stages could indeed be distinguished (66.58% interpretation ratio), which indicated differential expression signature. To investigate the expression patterns of the DEGs throughout the inflorescence development, a comparative transcriptomic analysis was performed using the DEseq2 software. A total of 14,579 significant DEGs were identified, wherein 7284 DEGs in MS1 vs. MS2, 12,257 DEGs in MS1 vs. MS3, and 7150 DEGs in MS2 vs. MS3 stages showed *p*-adjust value < 0.05 and |log2(Fold Change)| ≥ 1; 1799 DEGs were common among the three stages (Figure 3b,c). In the MS1 vs. MS2 comparison, 3770 DEGs were found to be upregulated, while 3514 were downregulated. In MS1 vs. MS3 comparison, 6369 DEGs were upregulated, while 5888 were down-regulated, which indicated that many upregulated differentially expressed genes (DEGs) occurred in the later flowering stages. In the MS2 vs. MS3 comparison, 3994 DEGs were upregulated, while 3156 were downregulated (Figure 3d). 

GO enrichment analysis of DEGs among the three groups was compared for the male inflorescence. The results suggested that the DEGs in the MS1 vs. MS2 stages were mainly enriched in the processes of the cell cycle (GO:0022402, BP), photosynthesis (GO:0015979, BP), photosystem (GO:0009521, CC), and stomatal complex development (GO:0010374, BP) (Appendix A). The DEGs between MS1 and MS3 were primarily associated with the chromosome (GO:0005694, CC), chromosomal part (GO:0044427, CC), and intrinsic components of the plasma membrane (GO:0031226, CC) (Appendix A). Additionally, the GO terms related to DEGs between MS2 and MS3 were enriched in hydrolase activity, hydrolyzing O-glycosyl compounds (GO:0004553, MF), DNA polymerase activity (GO:0034061, MF), hydrolase activity, acting on glycosyl bonds (GO:0016798, MF), and aspartic-type peptidase activity (GO:0070001, MF) (Appendix A). The KEGG enriched pathways related to DEGs in the male inflorescence at different periods are shown in Appendix A. The top four pathways in MS1 vs. MS2 included metabolic pathways (ko01100), biosynthesis of secondary metabolites (ko01110), phytohormone signal transduction (ko04075), and the MAPK signaling pathway in plants (ko04016) (Appendix A). In the MS1 vs. MS3 comparison, the significantly enriched pathways were metabolic pathways (ko01100), biosynthesis of secondary metabolites (ko01110), starch, and sucrose metabolism (ko00500), and inositol phosphate metabolism (ko00562) (Appendix A). The DEGs between MS2 and MS3 were related to the biosynthesis of secondary metabolites (ko01110), starch and sucrose metabolism (ko00500), pentose, and glucuronate interconversions (ko00040) (Appendix A). These results suggested that phytohormone signal transduction and glycometabolism play key roles during the development of the male flower of *J. mandshurica*.

### 2.4. Identification and Functional Enrichment of DEGs during Female Inflorescence Development

The expression patterns of genes also significantly changed during the development of the female inflorescence. In total, 49.14 million raw reads were obtained from nine cDNA libraries used for RNA-seq analysis. After multiple filtering steps, 46.50 million clean reads were obtained, having an average GC content of 45.88%. The total length of the clean bases was 7.75 Gb and the clean data from all the samples were greater than 7 Gb. The percentage of Q20 for all the samples was greater than 97%, with an average of 97.35%. In addition, the percentage of Q30 was greater than 92% in all samples and the average Q30 was 92.74% (Appendix A). Next, the clean data were aligned to the reference genome of *J. mandshurica* using the Hisat2 software. In total, the percentage of mapped reads from the nine samples ranged between 95.27% and 96.53%, accounting for an average of 96.31% of the clean data. The number of mapped unique reads was greater than 93%, while the percentage of multiple mapped reads in the reference genome was less than approximately 3% (Appendix A). In the present study, these results revealed a set of reliable RNA-seq data which could be utilized for subsequent analysis. To comprehensively understand the variability in the gene expression patterns during the development of a female flower, the DEGs from different comparison groups were identified. First, PCA was performed and used to detect the replicated samples at each stage; the results suggested good replicates and discrepancies between stages (Figure 4a). A total of 7188 DEGs were identified using the DEseq2 R package, including 2335 DEGs between FS1 and FS2 (980 upregulated and 1355 downregulated), 3295 DEGs (1459 upregulated and 1836 downregulated) between FS1 and FS3, and 1558 DEGs (938 upregulated and 620 downregulated) between FS2 and FS3 comparison groups (Figure 4b–d). 

To elucidate the molecular functions of DEGs obtained from different comparison groups, GO and KEGG enrichment analyses were further performed. GO enrichment analysis indicated that the GO terms related to secondary metabolic process (GO:0019748, BP), and heme-binding (GO:0020037, MF) were highly enriched in FS1 vs. FS2 (Appendix A). In the FS1 vs. FS3, the most significant terms were monooxygenase activity (GO:0004497, MF), heme-binding (GO:0020037, MF), and tetrapyrrole binding (GO:0046906, MF) (Appendix A). The DEGs in FS2 vs. FS3 were significantly enriched in ADP binding (GO:0043531, MF), aspartic-type endopeptidase activity (GO:0004190, MF), and aspartic-type peptidase activity (GO:0070001, MF) (Appendix A). Furthermore, we also investigated the major KEGG pathways to predict the specific biological functions of the DEGs. The most relevant pathways in FS1 vs. FS2 were involved in the biosynthesis of secondary metabolites (ko01110), plant–pathogen interactions (ko04626), phytohormone signal transduction (ko04075), and MAPK signaling pathway-plant (ko04016) (Appendix A). Interestingly, the DEGs obtained from the comparison of FS1 vs. FS3 were mainly related to the pathways similar to those in FS1 vs. FS2 (Appendix A) comparison groups. In particular, the top four pathways in FS2 vs. FS3 were involved in the biosynthesis of secondary metabolites (ko01110), followed by plant–pathogen interaction (ko04626), phenylpropanoid biosynthesis (ko00940), and flavonoid biosynthesis (ko00941) (Appendix A). Thus, pathways involved in phytohormone signal transduction and MAPK signaling pathway-plant were most prominent during the development of a female inflorescence.

### 2.5. Differentially Expressed Transcription Factors during the Transition to Flowering

Transcription factors (TFs) play an important role in gene expression and transcription and these are widely involved in the biological processes of plant growth and development, biosynthesis of secondary metabolites, as well as biotic and abiotic stresses. Herein, a total of 541 DEGs from the transition process in the male flower were identified relating to 43 TF families, while 247 DEGs related to the female flower were predicted to correspond to 33 TF families. In these key TFs related to the development of the male inflorescence, the largest number of TF families included ethylene response factor (ERF, 54), basic helix-loop-helix (bHLH, 39), Cys2/His2 zinc finger protein (C2H2, 39), v-myb avian myeloblastosis viral oncogene homolog (MYB_related, 17), and the combination of *NAM*, *ATAF*, and *CUC* genes (NAC, 35) (Appendix A). Furthermore, in addition to type I MADS (M-type MADS, 18), TFs including ERF (42), C2H2 (19), bHLH (17), and MYB_related (17) were also enriched during the development of the female flower (Appendix A). We, therefore, speculated that ERF, C2H2, bHLH, MYB_related, and MADS-box were crucial regulators of the processes during transition to flowering, and their biological functions were thus subjected to further analyses.

### 2.6. Functional Analysis of the Common DEGs between Male and Female Inflorescence during the Transition to Flowering

To decipher the molecular functions of the identified DEGs during the processes of transition to flowering in *J. mandshurica*, we specifically focused on the common DEGs between the developing male and female inflorescence. From the Venn diagram, it was clear that a total of 3241 DEGs were commonly shared during the transition process in male and female flowers (Figure 5a). Additionally, it was clear that among these DEGs, the number of total DEGs and upregulated DEGs were maximum in FS1 vs. FS3 (2288 total and 1009 upregulated) and MS1 vs. MS3 (2584 total and 1325 upregulated), thereby indicating an obvious functional differentiation of genes during the flowering process (Figure 5b). Based on these analyses, a total of 163 DEGs were related to 32 families of TF, among which, the largest TF family was that of ERF (20, 12.27%), followed by bHLH (13, 7.98%), MYB_related (13, 7.98%), C2H2 (11, 6.75%), M-type MADS (9, 5.52%), trihelix (8, 4.91%), NAC (7, 4.29%), SBP (7, 4.29%), Dof (6, 3.68%), and GRAS (6, 3.68%) (Figure 5c and Appendix A). Interestingly, the first five TF families were consistent with those identified during the development of the female inflorescence. All the common DEGs were divided into 54 level-two functional classifications terms including 23 biological processes, 14 molecular functions, and 17 cellular components. Among them, the first three terms corresponding to biological processes were cellular (GO:0009987), metabolic (GO:0008152), and single-organism (GO:0044699). Among the cellular component terms, the majority of genes were classified into the cell (GO:0005623), cell part (GO:0044464), and organelle (GO:0043226). Additionally, the terms related to molecular functions were mainly involved in binding (GO:0005488), catalytic activity (GO:0003824), and nucleic acid-binding TF activity (GO:0001071) (Figure 5d and Appendix A). GO and KEGG enrichment analyses were also performed to understand the biological functions of all the common DEGs. The results indicated that the most significantly enriched GO terms included the membrane part (GO:0016020), cell periphery (GO:0071944), integral components of the membrane (GO:0016021), and intrinsic components of the membrane (GO:0031224) (Figure 5e). In KEGG analysis, the common DEGs were related with three pathways of cellular processes, four terms of environmental information processing, 20 pathways of genetic information processing, 97 pathways of metabolism, and two pathways of organismal systems (Appendix A). In particular, these common DEGs were significantly (q-value < 0.05) enriched in the pathways of plant–pathogen interaction (ko04626), MAPK signaling pathway-plant (ko04016), biosynthesis of secondary metabolites (ko01110), phytohormone signal transduction (ko04075), phenylpropanoid biosynthesis (ko00940), and flavonoid biosynthesis (ko00941), which represented the major metabolic processes related to the flowering transition in *J. mandshurica*.

### 2.7. Recognizing DEGs Related to Floral Transition and Flower Development

Based on the annotation results using public databases (GO, KEGG, NR, Swissprot, Tremble, KOG, and Pfam) (Appendix A), these DEGs related to the flowering pathways were mainly classified into three, including photoperiod, floral organ formation, and flower development. The results showed that the flowering-related genes maintained differential expression patterns in various developmental stages during male and female flower development. Photoperiod is a crucial biological process for plants as they respond to the light-dark cycle pattern of day and night. In total, 16 DEGs were identified in the photoperiod pathways. The majority of genes were highly expressed in the FS1 and MS1 stages. In particular, three genes (*gene-Jman002G0255000*, *gene-Jman015G0155500*, and *gene-Jman001G0136900*) showed significantly high expression in the MS3 stage, and these specifically encoded the gibberellin 20 oxidase, AGAMOUS-like 15 (AGL 15), and CONSTANS-like 3 (CO3) (Figure 6a and Appendix A). In the pathways of floral organ formation, 21 key genes were identified, which were mainly involved in the biological process of floral organ senescence, floral organ morphogenesis, floral organ abscission, and floral organ development. Herein, we found that the expressions of five genes including *gene-Jman005G0044900*, *gene-Jman015G0132600*, *gene-Jman012G0034100*, *gene-Jman008G0031700*, and *gene-Jman014G0038100* were markedly upregulated during the FS2 and FS3 stages, and they were responsible for the BARELY ANY MERISTEM 1 (BAM1), YABBY 1-like, YABBY 1, BEL1-like homeodomain protein 9 (BLH9) and YABBY 4-like protein (Figure 6b). Additionally, two genes (*gene-Jman013G0095900* and *gene-Jman006G0073000*) encoding proteins of gibberellin receptor GID1B-like and AGAMOUS-like 42 (AGL 42) showed relatively high expression levels in the MS3 stage during the development of the male flower. Furthermore, 45 genes belonging to the flower development pathway were further identified and their expression patterns in different developmental periods of the female and male inflorescence were substantially different. As shown in Figure 6c, a total of seven genes, *gene-Jman006G0173300*, *gene-Jman015G0130400*, *gene-Jman002G0248700*, *gene-Jman003G0052400*, *gene-Jman004G0007500*, *gene-Jman006G0046000*, and *gene-Jman001G0387900*, encoding jasmonate ZIM domain (JAZ), patatin-like lipase (PTL), WD repeat-containing protein RUP2 (RUP2), Myb family transcription factor EFM, SOC1, CML 24, and AP2-like ethylene-responsive TF ANT protein, respectively, showed high expression in the FS2 and FS3 stages. Interestingly, five genes, *gene-Jman002G0255000*, *gene-Jman009G0035500*, *gene-Jman004G0024500*, *gene-Jman015G0028300*, and *gene-Jman006G0130100*, encoding gibberellin 20-oxidase, zeatin O-glucosyltransferase, tryptophan aminotransferase-related, pheophorbide and oxygenase and Zinc finger protein, and CONSTANS-like 1 (CO1) protein, respectively, were specifically expressed in the MS2 and MS3 stages, indicating positive regulation during the development of the male inflorescence.

### 2.8. Identification and Expression Analysis of Genes Involved in Biosynthesis and Reception of Phytohormones

In a previous study, floral organ formation and flower development in plants are affected by massive phytohormones. Herein, to further understand the functions of phytohormones for controlling the flower development in *J. mandshurica* in a coordinated manner, the DEGs related to phytohormone signaling pathways were identified, and the heatmap for their expressions is shown in Figure 7. In total, 42 genes involved in phytohormone pathways were obtained and these could be divided into seven groups according to the category, including ABA, CTK, ETH, GA, IAA, JA, and ZT. Among these genes, the expression of one JA-related gene (*gene-Jman009G0065500*), one CTK-related gene (*gene-Jman009G0035500*), one EHT-related gene (*gene-Jman002G0188500*), two GA-related genes (*gene-Jman004G0241300* and *gene-Jman002G0255000*), and two ZT-related genes (*gene-Jman008G0270700* and *gene-Jman002G0294300*) in the MS2 and MS3 stages were significantly higher relative to the MS1 stage. In particular, we found four genes including one JA-related gene (*gene-Jman016G0073600*), one GA-related gene (*gene-Jman013G0089200*), and two CTK-related genes (*gene-Jman012G0010100* and *gene-Jman002G0308500*) that consistently showed relatively high expression levels in the FS3 and MS3 stages, which indicated that they may play important roles in the flower development processes. Taken together, GA may occupy a dominant position in regulating the flower development in *J. mandshurica*.

### 2.9. DEGs Involved in MAPK Signaling Pathway

The MAPK cascade plays an important role in the regulation of plant growth and stress resistance. As stated previously, the DEGs obtained from KEGG enrichment analysis, in female and male inflorescence, were annotated to the MAPK signaling pathway (plant) (ko04016), which is complex. A total of 76 structural genes related to this pathway were identified, and the FPKM value of each gene was used to plot the heatmap (Figure 8). These genes were assigned to multiple signals, including ethylene, JA, and H_2_O_2_. Among these identified genes, most of them showed high expression in the female flower relative to the male flower. In the ethylene and ABA pathways, the expressions of two *BASIC CHITINASE* (*ChiB*) genes, one *SnRK2* gene, and *PYP/PYL* gene were found to be significantly higher in the MS3 stage as compared to the FS3 stage, which could promote the deference responses and stress adaptation during the development of the male flower. Furthermore, in the JA pathway, six *MYC2* genes and four *PR1* genes were identified, which could contribute to improving the wounding responses of the developing flower. The results implied that these genes played an important role in response to stress during floral transition and flower development in *J. mandshurica*.

### 2.10. DEGs Involved in Energy Metabolism during Flowering Process

In addition to the above-mentioned biological pathways, in the energy metabolism pathways, several genes were enriched, including starch and sucrose metabolism, glycolysis, pentose phosphate pathway, and galactose metabolism; further, many genes encoding some key enzymes were identified (Figure 9). In the present study, most of the genes encoding sucrose synthase (SUS) were downregulated or absent during flower development, while they maintain a high expression level in the initial stages of flower development. Furthermore, the genes involved in the pentose phosphate pathway were mostly inhibited, and only one gene encoding 6-phosphogluconolactonase was identified. In particular, most of the genes related to glucose metabolism were markedly upregulated during development, especially those encoding trehalose-phosphatase, with a high expression in the MS3 stage. The genes involved in galactose, melibiose, and fructose metabolism were highly expressed during the development of female flowers such as those encoding beta-fructofuranosidase. For starch metabolism, the number of genes encoding alpha-1,4 glucan phosphorylase (37 genes) was the largest, followed by alpha-amylase (26 genes), and most of them maintained relatively high expression levels at the late stage of flower development.

### 2.11. Co-Expression Network Analysis and Identification of MADS Genes

To identify the regulatory relationships between TFs, flowering-related, and phytohormone-related genes, the co-expression network was constructed using the homologous proteins in *Arabidopsis* in the STRING database; the network was further visualized using the Cytoscape software. In the first network (Figure 10a), the identified flowering-related genes and the top ten TF families relevant to the common DEGs were used to construct the network. The results showed that these TFs displayed strong interactions with the flowering-related genes. In particular, three genes including *gene-Jman001G0049900*, *gene-Jman013G0155600*, and *gene-Jman005G0221900* related to *SOC1/AGL20*, *ANT*, and *SVP*, respectively, showed strong interaction with TFs as compared to other genes in the present network, which implied that these may play an important role in regulating flower development in *J. mandshurica*. Additionally, we found that the M-type MADS TFs showed the strongest interaction with other genes, and were thus considered the hub regulators. In the second network (Figure 10b), the interaction between the TFs and phytohormone pathway-related genes was obtained. Indeed, the M-type MADS TFs showed high connectivity with other genes, which further supported their key functions during flower development. Additionally, two genes including *gene-Jman002G0294300* (cytokinin hydroxylase-like) and *gene-Jman001G0068100* (gibberellin 3-beta-dioxygenase 1), showed consistently high interaction with other genes. These results indicated that the M-type MADS TFs had a strong interaction with the genes involved in flower development, and thus, could be further considered the hub regulators.

In total, nine M-type MADS genes from the co-expression network were selected and their expression profiles were further investigated in female and male inflorescence development. As shown in Figure 10c, four M-type MADS genes including *gene-Jman004G0070700* (*AGL8*), *gene-Jman011G0145000* (*AGL9*), *gene-Jman009G0114300* (*GAG2*), and *gene-Jman015G0155500* (*AGL15*) were specifically and highly expressed in the FS2 and FS3 stages. Furthermore, two M-type MADS genes including *Jman004G0201100* (*AGL19*) and *Jman011G0145000* (*AGL9*) showed high expression levels in the MS2 and MS3 stages. In particular, three M-type MADS genes including *Jman009G0114300* (*GAG2*), *Jman005G0221900* (*SVP*), and *Jman006G0073000* (*AGL42*) displayed high expression levels in the MS3 stage, which indicated an important regulatory function in the late stages of male flower development (Figure 10d). In particular, *Jman011G0145000* (*AGL9*) displayed a similar expression pattern in the male and female flower development and was considered as the key regulator of flower development in *J. mandshurica.*

### 2.12. Validation of RNA-Seq Data by qRT-PCR

To verify the availability and accuracy of the RNA-seq data, a total of 16 candidate genes were selected and validated in qRT-PCR assays (Figure 11). Herein, eight DEGs were selected corresponding to the development of female flowers, while the other eight DEGs were obtained from the results related to the development of the male flower. This experiment was performed in the samples collected from female and male inflorescence development stages (FS1, FS2, FS3, MS1, MS2, and MS3). The primer sequences of candidate genes are listed in Appendix A. The results showed that the expression patterns of candidate genes in the qRT-PCR assay were consistent with those in the RNA-seq analysis.

## 3. Discussion

Flowers are important reproductive organs in plants. Flowering plays an important role in the life cycle of a plant, and it directly determines the quantity and quality of the fruit, thereby determining its commercial value. *J. mandshurica* is a medicinal and woody oil plant that is known for its specific effects on various tissues [34]. Understanding the processes and molecular mechanisms underlying flowering will contribute to improving the fruit yield and accelerate the commercial usage of *J. mandshurica*. In the present study, based on a high-quality reference genome, we comprehensively investigated the regulatory mechanisms of transition to female and male flowering, which is expected to provide a reference for reproduction genetics in *J. mandshurica* based on the available genetic information.

### 3.1. Identification of DEGs Involved in Flower Development in J. mandshurcia

Flowering in plants was mainly assigned to three key developmental processes according to the morphological and physiological characteristics, including flowering determination, flower evocation, and floral organ development. In general, some key hub genes, including *FT*, *SOC1*, and *CO*, form a complex flowering regulatory network to coordinate flowering in plants [12,35]. However, the functions of genes involved in flower development in *J. mandshurica*, remain unclear. In this study, several key genes related to photoperiod, GA pathway, floral pathway integrators, and flowering inhibitory factors were identified (Figure 12). In *A. thaliana*, *CONSTANS-like* (*CO*) genes are essential for regulating the flowering time [36], and its homologous genes are also found in many flowering plants including *Vigna radiata* [37], *Cannabis sativa* [38], *Brassica napus* [39] and *Populus trichocarpa* [40]. In particular, *CO* can activate the transcription of *FT*, resulting in the transfer of the FT protein from leaf phloem to the shoot apex meristem, which can promote plant flowering [39]. Xiao et al. have studied the specific functions of *CO* in *Phyllostachys violascens* and show that the overexpression of *PvCO1* in *Arabidopsis* can delay flowering by reducing the expression of the *FT* gene [41]. In this study, two *CO* genes (gene-*Jman001G0136900* and *gene-Jman006G0130100*) encoding *CO3* and *CO1* protein, respectively, were identified, and we found that the expressions of these two genes in the MS3 stage were markedly higher than those in the MS1 and MS2 stages, thus indicating a specific positive regulatory function for male inflorescence development (Figure 6c). *CO3* is a positive regulator for photomorphogenesis [42], and thus, we further speculated that it may also participate in the photoperiod-mediated flowering processes in *J. mandshurica*. In addition to the photoperiod pathway, the genes involved in GA biosynthesis and signal transduction also play key roles in the regulation of plant flowering through the integration of the floral pathway integrators such as *FT*, *SOC1*, and *LFY* [43]. DELLA proteins belonging to the GRAS gene family are key proteins, which are widely implicated in the GA signaling transduction [44,45]. Previous studies show that the DELLA protein, an inhibitor of the GA pathway, negatively regulates flowering in *A. thaliana* [25]. In particular, a total of five DELLA proteins are found in Arabidopsis, wherein *RGL1*, *RGA*, and *RGL2* are related to floral development [46,47]. Herein, one DELLA protein (*gene-Jman013G0095900*) was found to be highly expressed in the MS3 stage in the male flower; in contrast, it was down-regulated in FS3 as compared to the FS1 stage (Figure 6b). We reasonably speculate that this gene is a key specific regulator controlling the development of male inflorescence. A recent study reports that the DELLA protein can directly inhibit the expression of *CO* in the photoperiod pathway and co-regulate flowering induction in plants, consistent with the results of the present study [48]. Additionally, a gibberellin receptor (*GID1B*, *gene-Jman013G0095900*) was identified, which may combine with gibberellin and further promote the integration of *GID1* and *DELLA*, forming a GID1-GA-DELLA complex which can activate the ubiquitination of the DELLA protein. Furthermore, a *BAM1* gene (*gene-Jman005G0044900*) and three *YABBY* genes (*gene-Jman015G0132600*, *gene-Jman012G0034100*, and *gene-Jman014G0038100*) showed relatively high expression in the FS2 and FS3 stages as compared to the FS1 stage, implicating their major roles in the regulation of the development of female inflorescence. Taken together, these genes play a key role during the flowering transition process in *J. mandshurica*.

### 3.2. Underlying Functions of Phytohormone in Flower Development in J. mandshurcia

Phytohormones are important growth regulators, which are widely involved in the regulation of plant growth and stress responses, especially in the regulation of flowering [49,50,51]. However, the regulatory mechanisms underlying flowering in *J. mandshurica* remain ambiguous. Gibberellin plays an important role in plant growth and flower development [52]. In the present study, significant (*p* < 0.05) differences were observed in the GA content in different stages of male flower development, suggesting that it plays a crucial role in flower development in *J. mandshurica*. Additionally, a relatively high expression was observed in the FS1 and MS3 stage as compared to the other stages in flower development, thereby indicating differential functions of gibberellin in female and male inflorescence development in *J. mandshurica* (Figure 2). Gibberellin 20 oxidase (GA-20 oxidase) is a key enzyme for the biosynthesis and regulation of gibberellin [53]. In this study, two genes (*gene-Jman002G0255000* and *gene-Jman004G0241300*) encoding gibberellin 20 oxidase were identified and showed high expression levels in the MS2 and MS3 stages, thereby contributing to the accumulation of gibberellin (Figure 7). Upon increase in the gibberellin concentration, the DELLA protein is degraded through the ubiquitination pathway, which in turn promotes flowering [54]. Several studies show that ABA is related to flowering and affects the flowering time [55]. In *J. mandshurica*, ABA content also showed significant differences (*p* < 0.05). CTK is a key regulator and is involved in many biological processes throughout the life cycle of plants. In *Arabidopsis*, CTK activates the *TWIN SISTER OF FT* (*TSF*) to promote flowering [56]. Herein, two CTK-related genes (*gene-Jman012G0010100* and *gene-Jman002G0308500*) were highly expressed in the late stages of female and male flowers, thereby suggesting a significant regulatory function in flowering in *J. mandshurica*. Two genes (*gene-Jman002G0294300* and *gene-Jman001G0068100*) encoding cytokinin hydroxylase-like and gibberellin 3-beta-dioxygenase 1 protein showed strong interactions with other genes and these may be the hub genes in the hormone signaling pathway. 

### 3.3. Energy Metabolism during Flower Development

Carbohydrates not only serve as energy providers and carbon skeletons in the plant life cycle but are also important signaling molecules regulating plant growth and development [57]. Sugar metabolism plays an important role in the reproductive development of plants and contributes to the formation of the cell wall, pollen maturation, and pollen tube growth [58]. During flower development, sugar metabolism and transport are relatively complex and include multiple biological processes such as synthesis, transport, and metabolism [59,60]. SUS and sucrose invertase (INV) are the two main enzymes that contribute to sucrose metabolism. In the present study, most *SUS* genes showed high expression levels at the early stages of inflorescence development, suggesting that they may largely contribute to flower bud differentiation (Figure 9). Furthermore, in the starch metabolism pathway, the number of structural genes was the highest and they were highly expressed at the late stages of flower development, thereby implying rapid starch metabolism in the development of female and male inflorescence. Subsequently, these starch molecules can be further converted into many soluble sugars such as glucose and galactose for supporting the energy demand during inflorescence development.

### 3.4. MADS-Box Genes Contribute to the Flowering of J. mandshurica

Floral organ transition and flower development are complex processes regulated by many TFs, especially the MADS-box gene family [61,62,63]. At present, multiple MADS-box genes have been identified in many plants, and their functions have also been verified in *Medicago sativa* [64], *Ipomoea batatas* [65], *Adonis amurensis* [66], *Nelumbo nucifera* [67] and *Phyllostachys edulis* [68]. However, the potential regulatory functions of MADS-box genes during floral organ development in *J. mandshurica* remain unclear. The I-type (M-type) MADS-box genes are mostly associated with multiple reproductory processes, including embryogenesis, gametogenesis, flower development, and seed formation [69]. Using the high-quality RNA-seq data, several M-type MADS-box genes were identified in the present study and they contributed to the regulation of flowering in *J. mandshurica*. *SOC1* is a key integrator and is relatively conserved in the plant flowering pathways [70]. In *Arabidopsis*, *SOC1* belonging to the MADS-box gene family is considered a key regulator and can integrate multiple signaling pathways including in photoperiod, autonomous floral induction, and vernalization; it interacts with flowering-related genes to control the flowering processes such as flowering time and floral meristem differentiation [71]. In *Bambusa oldhamii*, *BoMADS50,* a *SOC1-like gene,* is considered a crucial regulator that positively controls the flowering process [72]. Similarly, a previous study shows that the overexpression of *MtSOCl* can significantly promote flowering and contributes to the elongation of the primary stem in *Medicago truncatula* [73]. In this study, the *gene-Jman004G0007500* encoding *SOC1* protein was progressively upregulated from FS1 to FS2 stage, while it was downregulated from MS1 to MS2 stage, which implied that *SOC1* may positively mediate transition in the female flowering process. Additionally, this gene showed a strong interaction with the other genes involved in flower development in the co-expression network (Figure 10a), which further supported the results obtained from the previous study. Furthermore, our results also suggested the important role of *SHORT VEGETATIVE PHASE* (*SVP*) genes in *J. mandshurica* during flower development. *SVP* is a well-known member of the MADS-box gene family that negatively regulates the early stages of floral transition in *Arabidopsis* [74,75]. In particular, the expression of the *SOC1* gene is inhibited by *SVP* during the flowering process, suggesting an interaction between them [76]. Previous studies show that the overexpression of *MtSVP1* can delay flowering in *Arabidopsis*, while it only changes the floral organ characteristics in *Medicago truncatula* [77]. In this study, we found that one *SVP* (*Jman005G0221900*) gene showed differential expression pattern in female and male inflorescence development (Figure 10c,d), and was highly expressed in the early stages of the female flower development, while showed opposite trends in male flower, indicating that *SVP* may play a key positive regulatory role in the development of the female flowers of *J. mandshurica* while exerting negative regulation in the development of male flowers. Thus, this gene may be a potential candidate for further studies on flower development in *J. mandshurica.*

In addition to *SOC1* and *SVP*, numerous *AGAMOUS-like* (*AGL*) genes such as *AGL8*, *AGL9*, *AGL15*, and *AGL42* were also identified in *J. mandshurica* flower developmental stages through RNA-seq analysis. *AGLs* belonging to the MADS-box TF family are important floral organ identification genes, primarily responsible for flowering transition repression in many flowering plants, including *Poncirus trifoliata* [78], soybean [79], *A. thaliana* [80], *Populus trichocarpa* [81] and *Brassica juncea* [82]. At present, the potential functions of AGL genes have been confirmed in *Arabidopsis*, and most of these show specific regulation of the floral transition process [83], for example, the overexpression of *AGL17* in the early stages of flowering, while the functional inhibition of this gene causes the delay in flowering time [84]. In our study, two *AGL* genes, *JmAGL8* and *JmAGL9*, were detected at high levels in the late stages of female flower development, while four *AGL* genes, *JmAGL9*, *JmAGL15*, *JmAGL19*, and *JmAGL42,* showed a consistently high expression at the late stages of male flower development. These results suggested that *AGL* genes play a role in the floral organs and flowering processes in *J. mandshurica*. Interestingly, the expression of *JmAGL9* was consistently high at the late stages of female and male inflorescence development. It is necessary to further investigate the roles of *JmAGL9* in the flower development of *J. mandshurica*. From the co-expression network, one *SOC1/AGL20* was identified and showed strong interactions with TFs as compared to other genes. In particular, it was differentially expressed and showed relatively high expression in the early stages of male flower development. Taken together, these results suggested that the *AGL* genes regulated flowering in *J. mandshurica*.

## 4. Materials and Methods

### 4.1. Plant Materials and RNA Preparation 

*J. mandshurica* mainly distributed in Northeast China and has unique flowering phenology and reproductive feature. The experimental materials used in this study were obtained from the campus of Northeast Forestry University in May 2021 in Harbin, Heilongjiang province, China (45°43′6.53″ N, 126°37′57.28″ E). In particular, the adult trees used for *J. mandshurica* whole-genome sequencing were selected for collecting the male/female flower samples during the flowering stage. Before flower natural pollination, a total of six samples from the stages of male (MS1, MS2, and MS3) and female (FS1, FS2, and FS3) inflorescence were collected and defined as early-stage, mid-stage and late developmental stage, and the photographs of different developmental period was shown in Figure 1. For each developmental stage of male/female flower, a total of 5 inflorescences with the same development status were considered as one mixed sample. The samples for each stage have three biological repeats. The male/female flower samples separated from inflorescence were rapidly freezed by liquid nitrogen and then stored at −80 °C refrigerator for subsequent RNA isolation, library construction and transcriptome sequencing. Total RNA was extracted from the flower samples using the TRIzol reagent (Invitrogen) based on the manufacturer’s protocol. After that, the concentration and purity of total RNA was further assessed by the Qubit 3.0 Fluorimeter (Life Technologies, Carlsbad, CA, USA) and NanoDrop 2000 spectrophotometer (NanoDrop Technologies, Wilmington, DE, USA), respectively. The integrity and quality were detected using 0.8% agarose gel electrophoresis. After obtain the high-quality RNA, the cDNA library was constructed and used for RNA-seq.

### 4.2. Measurement of Hormone Content

To observe the concentrations and change of endogenous hormones during developmental flower, a total eight hormones for each stage were measured in present study. The content of ethylene (ETH), abscisic acid (ABA), cytokinin (CTK), auxin (IAA), jasmonic acid (JA), Zeatin (ZT), brassinosteroids (BR) and gibberellins (GA) were obtained using enzyme linked immunosorbent assay (ELISA) and high-performance liquid chromatography (HPLC) by the Shanghai Enzymatic Biotechnology Company Ltd. (Shanghai, China). All data from hormone content were analyzed by the variance (ANOVA) analysis. In particular, the differences analysis were performed using the IBM SPSS Statistics v25.0 software with the Student–Newman–Keuls multiple range test. Bars with different lowercase letters are significantly different (*p* < 0.05).

### 4.3. RNA-Seq Library Construction and Sequencing

Approximately 3 μg total RNA of each sample was used as raw materials to constructure the cDNA library. The amplification and synthesis of 18 cDNA libraries from three stages of male/female flower were obtained and further utilized for sequencing. In particular, the RNA-seq was performed on an Illumina HiSeq sequencing platform (Illumina, San Diego, CA, USA) with 150 bp long paired-end reads, and then the obtained raw data were used for subsequent analysis. Transcriptome sequencing data are available in the SRA database of the National Center for Biotechnology Information (NCBI) under the accession number of PRJNA805360.

### 4.4. Bioinformatics Analysis of RNA-Seq Data

To ensure and obtain the high-quality clean data for further bioinformatics analysis, the raw data were filtered to remove adapter sequences and low-quality bases using Fastp software (version 0.12). Meanwhile, the Q20, Q30 and GC content were obtained and utilized to evaluate the sequencing quality. The reference fasta sequences and annotation files of chromosomal-level *J. mandshurica* genome was obtained from the Genome Warehouse in National Genomics Data Center (NGDC) (https://ngdc.cncb.ac.cn/) (accessed on 11 April 2022) under accession number PRJCA006358. The index of reference genome was constructed using hisat2-build tools, and then the obtained clean data were further mapped to the *J. mandshurica* reference genome using the HISAT2 software (version 2.1.0) [85]. Then, the quantitation of gene expression was performed using the featureCounts software, and the obtained raw counts matrix was used to calculate the Fragments per kilobase of transcript per million fragments mapped reads (FPKM) value as the gene expression level in different samples [86]. 

### 4.5. Differential Gene Expression Analysis and GO Enrichment Analysis

Identification of the differentially expressed genes (DEGs) at different developmental periods of male/female flowers was performed using the DESeq2 R package (1.20.0) based on the raw count matrix as the data input [87]. In particular, through DESeq2, the differences between comparative groups were determined and obtained using the negative binomial distribution. At the same time, using the Benjamini and Hochberg’s false discovery rate (FDR), the obtained *p*-value was further adjusted and used for digging DEGs with the following parameters: *p*-adjust value < 0.05 and |log_2_Fold Change (FC)| ≥ 1. 

To identify the specific function of DEGs of different comparison groups, the Wallenius non-central hyper-geometric distribution, GOseq R package was used to implement the Gene Ontology (GO, http://geneontology.org/) (accessed on 11 April 2022) enrichment analysis [88]. Furthermore, the DEGs enriched in Kyoto Encyclopedia of Genes and Genomes (KEGG, http://www.kegg.jp/) (accessed on 1 November 2021) pathways were identified using the KOBAS 2.0 software [89]. The visualization analysis of enrichment results was implemented on the Omicsmart platform (Genedenovo Biotechnology Co., Ltd., Guangzhou, China; https://www.omicsmart.com/) (accessed on 15 November 2021). After that, the transcription factors (TFs) from DEGs were captured and predicted by integrating the iTAK software (https://github.com/kentnf/iTAK) (accessed on 15 November 2021) and PlantTFDB database (http://planttfdb.gao-lab.org/) (accessed on 15 November 2021).

### 4.6. Interaction Network Construction

To obtain the interaction relationship of the candidate proteins, protein-protein. interaction network (PPI) was constructed on STRING protein interaction database (http://string-db.org) (accessed on 16 November 2021) with default parameters, in which the candidate protein sets in this study were directly mapped to homologous proteins in *A. thaliana*. According to the interaction relationship, the co-expression network was obtained. After that, the protein interaction results were imported into Cytoscape software to obtain the final regulatory network.

### 4.7. Verifying Gene Expression Levels by qRT-PCR

To evaluate the accuracy and reliability, the expression pattern of twelve genes (six from male flower, six from female flower) in present study were selected and used for Quantitative real-time polymerase chain reaction (qRT-PCR) with three technical replicates each reaction. The total RNA from different developmental flower samples was extracted and obtained using the plant total RNA isolation kit (TaKaRa, Kyoto, Japan). Then, approximately 1 μg total RNA was selected for cDNA amplification and synthesis according to the manufacturer’s instructions of cDNA Synthesis Kit (Takara, Kyoto, Japan). The qRT-PCR test was performed on an ABI 7500 RT PCR system. The primers used in this study were designed with an online website tool (https://sg.idtdna.com/scitools/Applications/RealTimePCR/default.aspx) (accessed on 20 December 2021), and the 18S-RNA was utilized as reference gene. The PCR reaction was implemented with the addition of 10 µL of 2x SYBR Premix Ex Taq, followed by 6 µL of double-distilled water (ddH_2_O), 0.8 µL of upstream and downstream primers (10 μmol/L), 2 µL of cDNA template, and 0.4 µL of ROX reference dye in 20 µL reaction mixture. The PCR reaction was at 94 °C for 30 s, 45 cycles of 94 °C for 5 s, 60 °C for 35 s, 95 °C for 15 s, 60 °C for 1 min, followed by 95 °C for 15 s. Finally, according to the 2^−ΔΔCT^ method, the relative expression level each gene in samples was calculated and obtained.

## Figures and Tables

**Figure 1 ijms-23-05433-f001:**
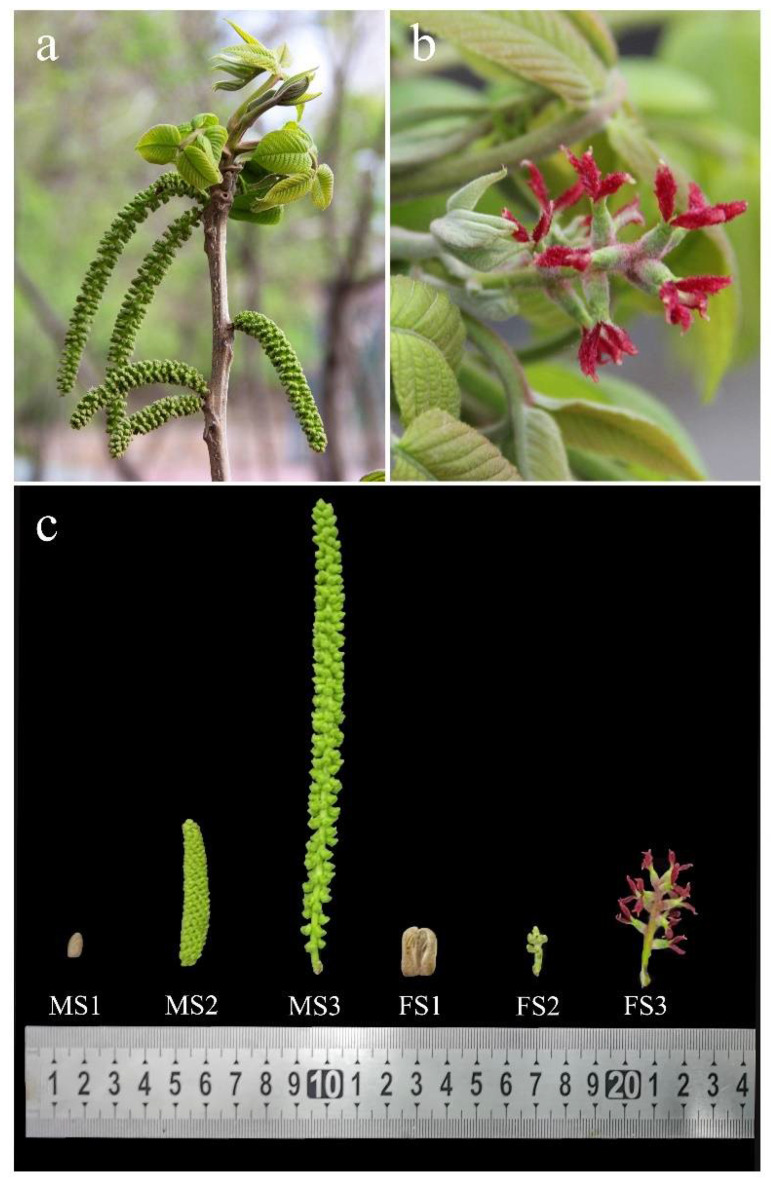
Morphological characteristics of male/female inflorescence of *Juglans mandshurica*. (**a**) male inflorescence; (**b**) female inflorescence; (**c**) the three key developmental stages of male (MS1, MS2, and MS3) and female (FS1, FS2, and FS3) inflorescence.

**Figure 2 ijms-23-05433-f002:**
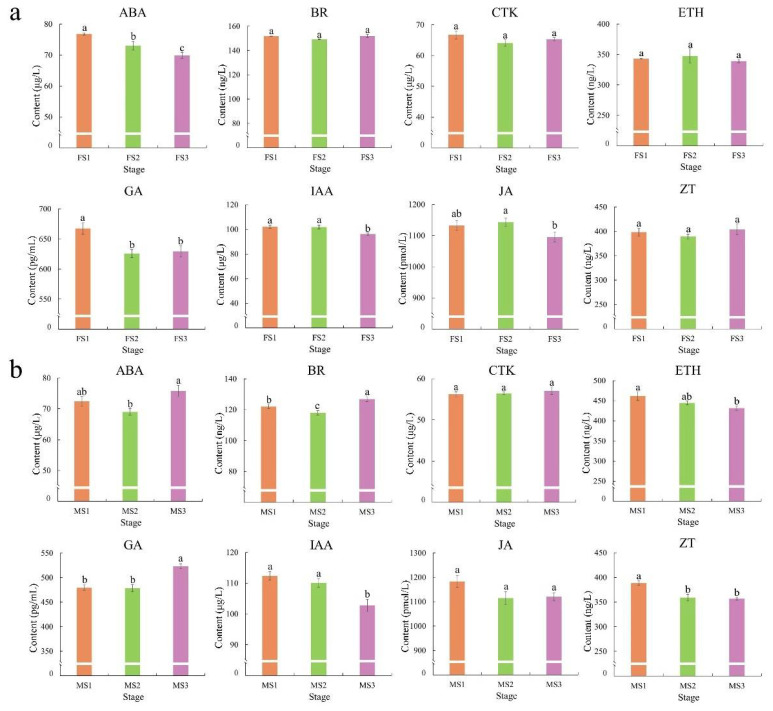
The phytohormones content during the flowering transition process. (**a**) female flower; (**b**) male flower. The *x*-axis represents the developmental stages of female and male, and the *y*-axis indicates the phytohormones content obtained by ELISA and HPLC. The differences analysis was performed using the IBM SPSS Statistics v25.0 software with the Student–Newman–Keuls multiple range test; Error bars represent the SD of the means at n = 3; Bars with different lowercase letters are significantly different (*p* < 0.05).

**Figure 3 ijms-23-05433-f003:**
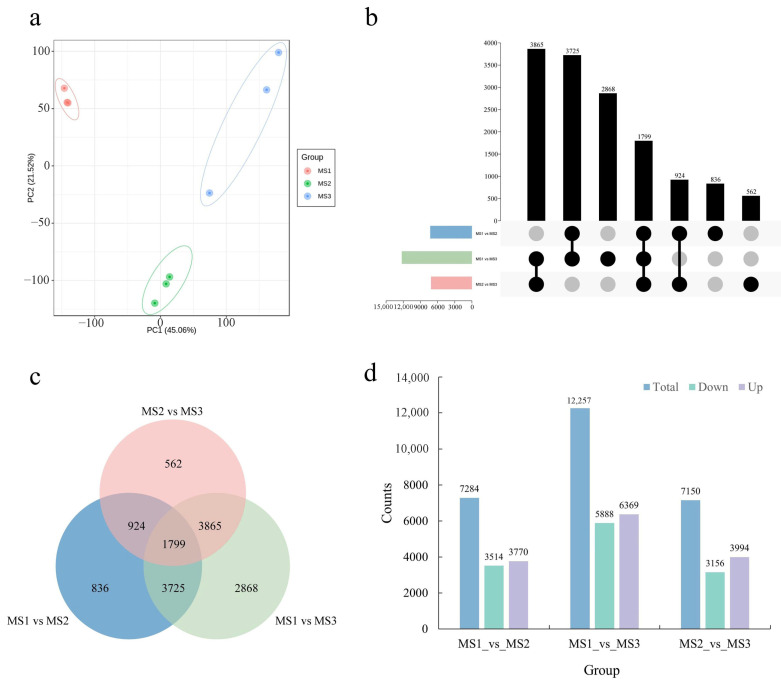
Identification and functional enrichment of differentially expressed genes at the male flowering transition process. (**a**) PCA score plot of expression profiles from different samples; (**b**) Upset diagram of DEGs in different comparison groups; (**c**) Venn diagram of DEGs in different comparison groups; (**d**) the statistical analysis of up-regulated and down-regulated DEGs among different comparison groups.

**Figure 4 ijms-23-05433-f004:**
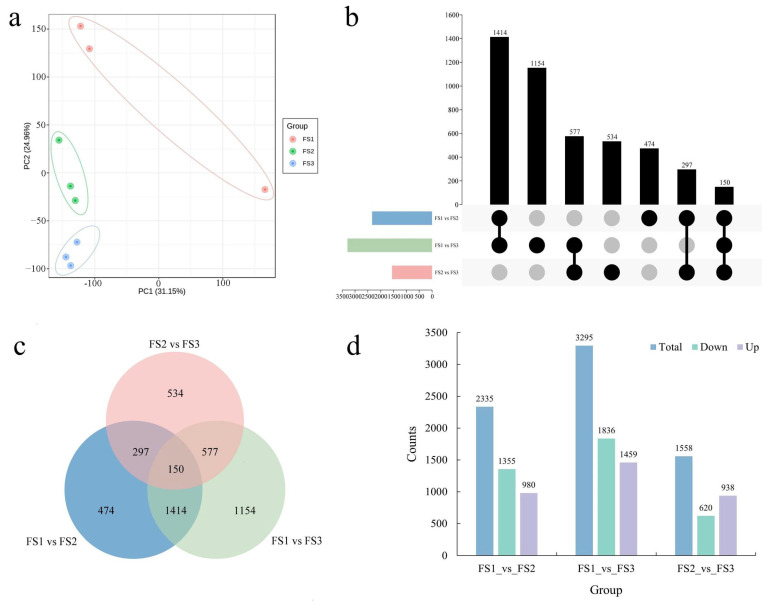
Identification and functional enrichment of differentially expressed genes at the female flowering transition process. (**a**) PCA score plot of expression profiles from different samples; (**b**) Upset diagram of DEGs in different comparison groups; (**c**) Venn diagram of DEGs in different comparison groups; (**d**) the statistical analysis of up-regulated and down-regulated DEGs among different comparison groups.

**Figure 5 ijms-23-05433-f005:**
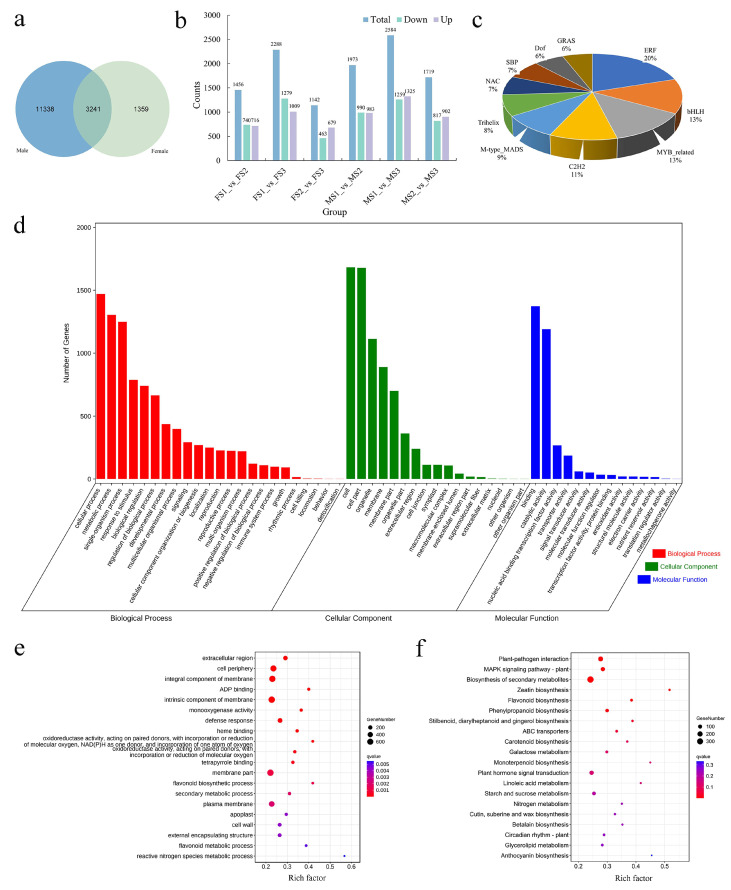
Preliminary analysis of the biological function of the common DEGs during the developmental male and female flower. (**a**) Venn diagram of DEGs obtained from male and female flower; (**b**) the statistical analysis of total, up-regulated and down-regulated DEGs in different comparison groups; (**c**) the pie chart displaying the first ten TFs in the common DEGs; (**d**) level-2 GO functional classifications of all the common DEGs; (**e**) GO enrichment analysis for the common DEGs; (**f**) KEGG enrichment analysis for the common DEGs.

**Figure 6 ijms-23-05433-f006:**
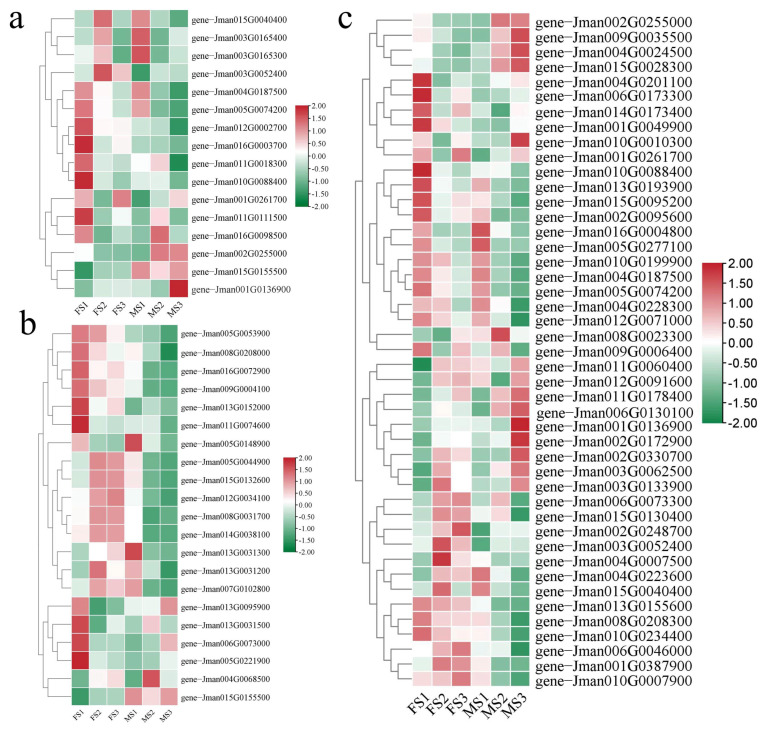
Identification of floral transition and flowering-related genes. (**a**) DEGs involved in photoperiod pathway; (**b**) DEGs involved in floral organ transition pathway; (**c**) DEGs involved in flower development pathway. The color scale from green to red indicates the expression value from low to high.

**Figure 7 ijms-23-05433-f007:**
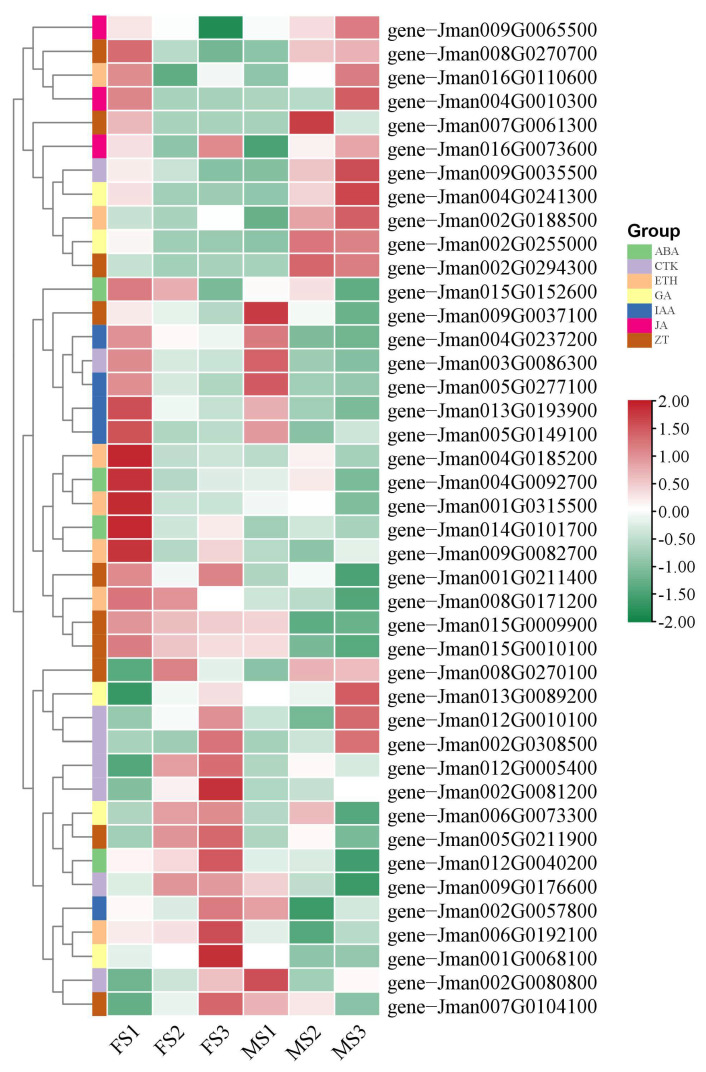
Identification and expression analysis of phytohormone during flower development. The color scale from green to red indicates the expression value from low to high.

**Figure 8 ijms-23-05433-f008:**
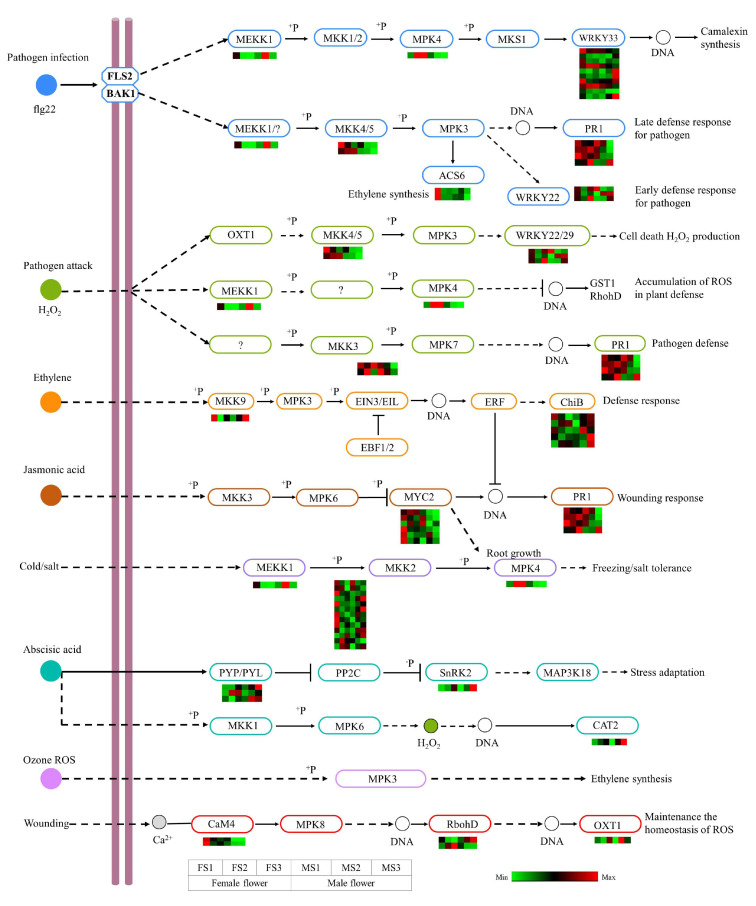
Analysis of DEGs related to the MAPK signaling pathway. The color scale from Min (green) to Max (red) refer to the expression value from low to high. ROS, reactive oxygen species.

**Figure 9 ijms-23-05433-f009:**
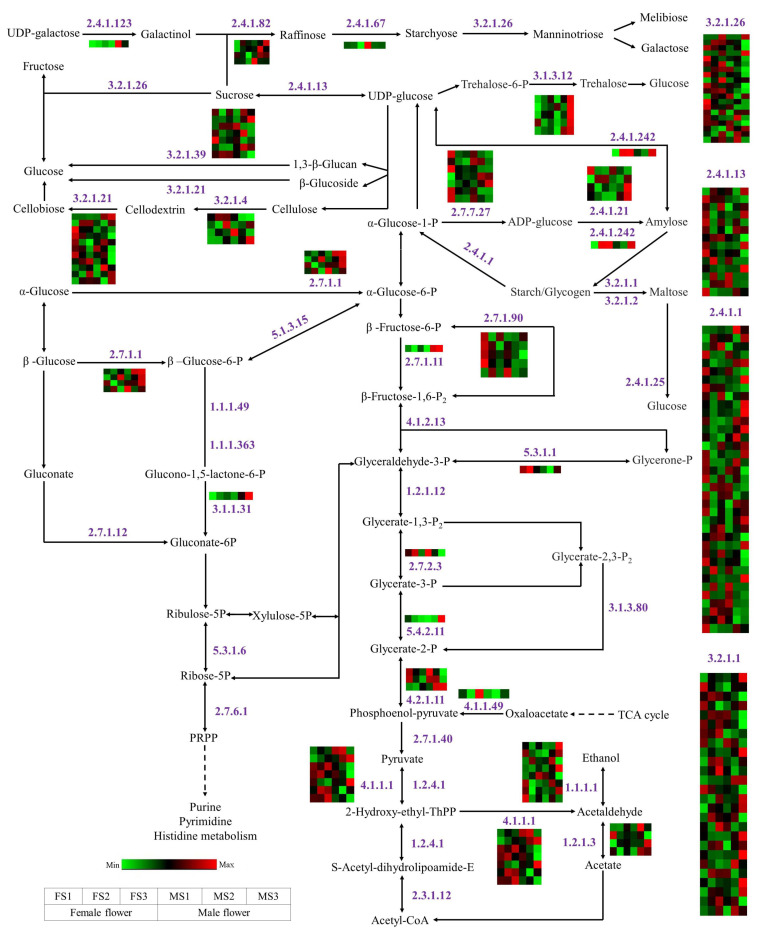
The genes related to energy metabolism during flower development in *J. mandshurica*. The number next to the arrow indicates the enzyme number. The color scale from Min (green) to Max (red) refer to the expression value from low to high. The box from left to right represents the expression level from FS1, FS2, FS3, MS1, MS2, and MS3, respectively. EC 2.4.1.123, inositol 3-alpha-galactosyltransferase; 2.4.1.82, galactinol-sucrose galactosyltransferase; 2.4.1.67, galactinol-raffinose galactosyltransferase; 3.2.1.26, beta-fructofuranosidase; 2.4.1.13, sucrose synthase; 3.1.3.12, trehalose-phosphatase; 3.2.1.39, glucan endo-1,3-beta-D-glucosidase; 3.2.1.21, beta-glucosidase; 3.2.1.4, cellulase; 2.7.1.1, hexokinase; 1.1.1.49, glucose-6-phosphate dehydrogenase (NADP+); 5.1.3.15, glucose-6-phosphate 1-epimerase; 1.1.1.363, glucose-6-phosphate dehydrogenase [NAD(P)+]; 3.1.1.31, 6-phosphogluconolactonase; 2.7.1.12, gluconokinase; 5.3.1.6, ribose-5-phosphate isomerase; 2.7.6.1, ribose-phosphate diphosphokinase; 2.4.1.242, NDP-glucose-starch glucosyltransferase; 2.7.7.27, glucose-1-phosphate adenylyltransferase; 2.4.1.21, starch synthase; 2.4.1.1, glycogen phosphorylase; 3.2.1.1, alpha-amylase; 3.2.1.2, beta-amylase; 2.4.1.25, 4-alpha-glucanotransferase; 2.7.1.90, diphosphate-fructose-6-phosphate 1-phosphotransferase; 2.7.1.11, 6-phosphofructokinase; 4.1.2.13, fructose-bisphosphate aldolase; 5.3.1.1, triose-phosphate isomerase; 1.2.1.12, glyceraldehyde-3-phosphate dehydrogenase; 2.7.2.3, phosphoglycerate kinase; 5.4.2.11, phosphoglycerate mutase; 3.1.3.80, 2,3-bisphosphoglycerate 3-phosphatase; 4.2.1.11, phosphopyruvate hydratase; 4.1.1.49, phosphoenolpyruvate carboxykinase; 2.7.1.40, pyruvate kinase; 4.1.1.1, pyruvate decarboxylase; 1.2.4.1, pyruvate dehydrogenase; 1.2.4.1, pyruvate dehydrogenase (acetyl-transferring); 2.3.1.12, dihydrolipoyllysine-residue acetyltransferase; 1.1.1.1, alcohol dehydrogenase; 1.2.1.3, aldehyde dehydrogenase (NAD+).

**Figure 10 ijms-23-05433-f010:**
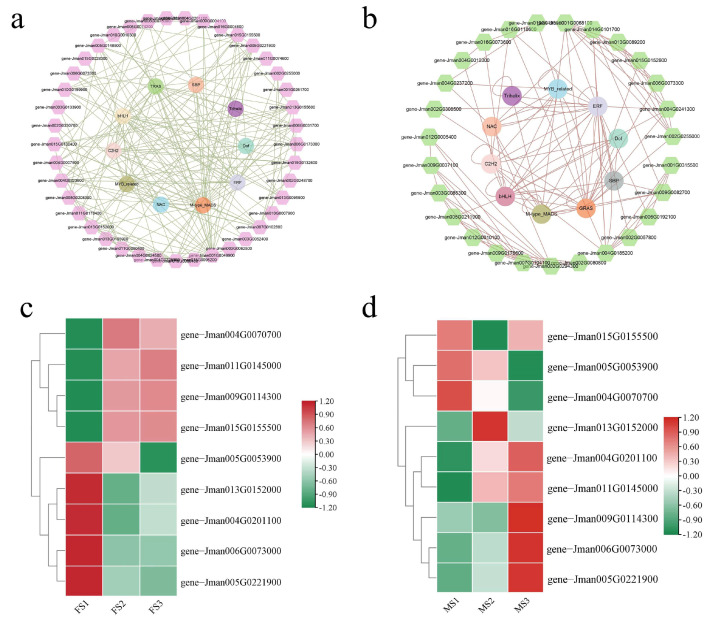
Co-expression network of identified TFs and genes, and the expression pattern of MADS-box genes. (**a**) the co-expression network between TFs and the floral transition and flowering-related genes; (**b**) the co-expression network between TFs and the phytohormone-related genes; (**c**) the expression level of M-type MADS during female flower development; (**d**) the expression level of M-type MADS during male flower development.

**Figure 11 ijms-23-05433-f011:**
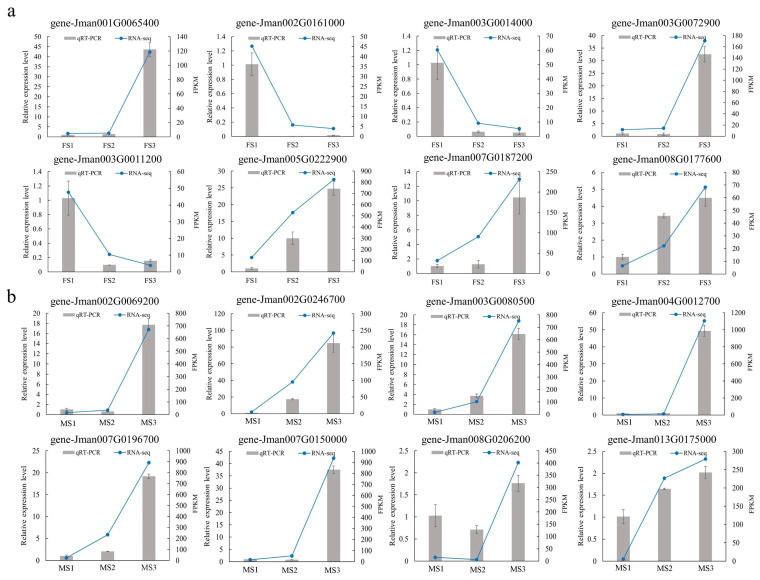
RT-qPCR verification of expression level of 16 DEGs identified by RNA sequencing. (**a**) RT-qPCR verification of eight DEGs during female development; (**b**) RT-qPCR verification of eight DEGs during male development. The *Y*-axis on the left indicates the relative gene expression levels (2^−−ΔΔCt^) analyzed by qRT-PCR, while the *Y*-axis on the right represents the FPKM value obtained by RNA-seq. The *X*-axis represents the different developmental stage samples.

**Figure 12 ijms-23-05433-f012:**
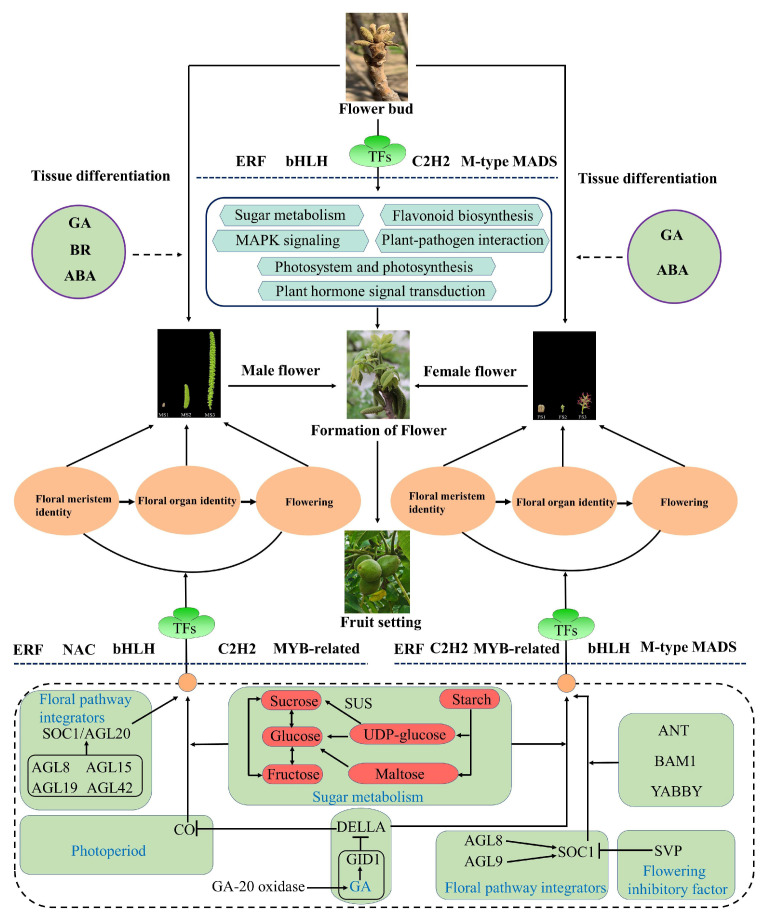
Schematic diagram of the regulatory mechanism for inflorescence development in *J. mandshurica*.

## Data Availability

All RNA-seq reads were deposited at Sequence Read Archive (SRA) database at NCBI with the accession number of PRJNA805360.

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
