# Peer review of "Characterization of Phytohormones and Transcriptomic Profiling of the Female and Male Inflorescence Development in Manchurian Walnut (Juglans mandshurica Maxim.)"

_ijms, 2022, doi:10.3390/ijms23105433_

Round 1

Reviewer 1 Report

The reviewed manuscript contains the results of a huge survey on regulation of floral ontogeny in Juglans mandshurica. This paper is generally well written. I have only two small concerns about this work.
1. In Juglandaceae, flowers are relatively minute, especially male ones. There is no commonly accepted agreement on a boarder between solitary flower and lateral inflorescence in this family. Female flowers are better discernible. However, almost everywhere, when writing about male/female 'flowers', authors actually mean inflorescences. I have changed these terms in some places but strongly recommend to check the rest of the text for this purpose.
2.  Some of figures (especially Figs. 3 and 4) are very huge and complicated and will be hardly readable even in a better resolution. I suggest either to split each of them into several smaller images or move some of plots to supplements.
I also strongly recommend authors to check all statements about genes putatively involved in glycogen metabolism, as plants do not have glycogen. Probably it will be more accurate to rephrase this.
Some of minor comments are available in a paper file (see attached). None of them are decisive, so I undoubtedly recommend this paper to be accepted for publication in the IJMS, after considering reviewer's suggestions.

Author Response

Thank you for your letter again and for the reviewers’ comments concerning our manuscript entitled “Characterization of phytohormones and transcriptomic profiling of the female and male flower development in Manchurian walnut (Juglans mandshurica Maxim.)” (ijms-1701820). Those comments are all valuable and very helpful for revising and improving the quality our paper. We have studied all comments carefully and have made corrections which we hope meet the standard of your highly esteemed journal. The main corrections in the paper and the responses to the reviewers’ comments are given below:

(1) In Juglandaceae, flowers are relatively minute, especially male ones. There is no commonly accepted agreement on a boarder between solitary flower and lateral inflorescence in this family. Female flowers are better discernible. However, almost everywhere, when writing about male/female 'flowers', authors actually mean inflorescences. I have changed these terms in some places but strongly recommend to check the rest of the text for this purpose.

Response: Thanks for your comments. The related sentences have been corrected in the whole manuscript.

(2) Some of figures (especially Figs. 3 and 4) are very huge and complicated and will be hardly readable even in a better resolution. I suggest either to split each of them into several smaller images or move some of plots to supplements.

Response: The Figure 3 and Figure 4 have been corrected.

(3) I also strongly recommend authors to check all statements about genes putatively involved in glycogen metabolism, as plants do not have glycogen. Probably it will be more accurate to rephrase this.

Response: The sentence has been corrected. See line 429.

(4) Some of minor comments are available in a paper file (see attached). None of them are decisive, so I undoubtedly recommend this paper to be accepted for publication in the IJMS, after considering reviewer's suggestions.

Response: Thanks for your comments. The related contents have been corrected in the whole manuscript.

Reviewer 2 Report

In this paper, the authors analyzed phytohormone measurements and transcriptomic analysis during flower development om Juglans mandshurica. However, the main mechanism has been explored in Arabidoopsis thaliana. The reviewer could not find the novelty of the research because the authors just conducted similar works in J. mandshurica. What's new? In addition, it was unclear why J. mandshurica was used in this study (poorly described in abstract, but very week). Furthermore, it was very difficult to understand what figures show because the letters were too small (or because of low resolution).

Minor points

1) Keywords. J. mandshurica and flower development were used in both the title and the keyword. The reviewer recommends using other words which was not used in the title. Sugar metabolism. This topic is a few remarked in discussion. Is this appropriate keyword?

2) p.3,L.121-136. Which figures did the authors explain in the paragraph? No citation of figures.

3) L.154 ETH, IAA, and ZT showed high expressions? Is this gene expression data? Figure 2 shows phytohormone levels. ETH, IAA, and ZT showed higher contents or levels?

4) L.573 'high expression' is not correct. Which phytohormone level was higher in the FS2? GA? If the authors cited gene expression data, figure number is wrong (not Figure 2). In Figure 2, GA of the FS2 is lower than FS1. I am confused.

5) L.585 What is the role of ABA during flower development in J. mandshurica?

6) L.593-595 Taken together... Everyone knows it.

7) PCA score plot in Fig. 3a MS3 and 4a FS1. One plot among three samples is far away from others. Are they really the same group?

8) No description on statistical analysis in the Materials and Methods section.

9) Cite appropriate figure number in the Discussion section.

10) Spell out abbreviations when the authors use at the first time. (ex. L.149, L.278-279)

11) FLOWERING LOCUS C (DELLA)? -> FLOWERING LOCUS C (FLC)

12) Figure 11b gene-Jman007G0196700. MS3 is lacking in the X-axis.

13) L.144 J. mandshurica should be written in italic.

14) L.759 and 761 TaKaRa, Beijing, China? Takara, Kyoto, Japan? Are they different company? It is strange for me.

15)Figure 7. Phytohormone name can be shown as abbreviations.

16) The reviewer highly recommends that English grammar of the text is checked by native specialist before submitting to the journal.

Author Response

Thank you for your letter again and for the reviewers’ comments concerning our manuscript entitled “Characterization of phytohormones and transcriptomic profiling of the female and male flower development in Manchurian walnut (Juglans mandshurica Maxim.)” (ijms-1701820). Those comments are all valuable and very helpful for revising and improving the quality our paper. We have studied all comments carefully and have made corrections which we hope meet the standard of your highly esteemed journal. The main corrections in the paper and the responses to the reviewers’ comments are given below:

(1) it was unclear why J. mandshurica was used in this study (poorly described in abstract, but very week).

Response: The sentences have been corrected in Abstract. See line 14-18.

(2) Furthermore, it was very difficult to understand what figures show because the letters were too small (or because of low resolution).

Response: The Figures have been corrected. See Figure 3 and Figure 4.

(3) Keywords. J. mandshurica and flower development were used in both the title and the keyword. The reviewer recommends using other words which was not used in the title. Sugar metabolism. This topic is a few remarked in discussion. Is this appropriate keyword?

Response: The keywords have been corrected. See line 33.

(4) p.3,L.121-136. Which figures did the authors explain in the paragraph? No citation of figures.

Response: The citation of figures has been added to the paragraph. See line 137.

(5) L.154 ETH, IAA, and ZT showed high expressions? Is this gene expression data? Figure 2 shows phytohormone levels. ETH, IAA, and ZT showed higher contents or levels?

Response: It is our mistake. The content of phytohormones changed during the female and male inflorescence development. The sentence has been corrected. See line 152 and 155.

(6) L.573 'high expression' is not correct. Which phytohormone level was higher in the FS2? GA? If the authors cited gene expression data, figure number is wrong (not Figure 2). In Figure 2, GA of the FS2 is lower than FS1. I am confused.

Response: The sentence has been corrected. See line 576.

(7) L.585 What is the role of ABA during flower development in J. mandshurica?

Response: The sentence has been deleted.

(8) L.593-595 Taken together... Everyone knows it.

Response: The sentence has been deleted.

(9) PCA score plot in Fig. 3a MS3 and 4a FS1. One plot among three samples is far away from others. Are they really the same group?

Response: We confirmed that they belong to the same group.

(10) No description on statistical analysis in the Materials and Methods section.

Response: The related description has been added to the manuscript. See line 703-707.

(11) Cite appropriate figure number in the Discussion section.

Response: The Figure citations have been added to the Discussion section.

(12) Spell out abbreviations when the authors use at the first time. (ex. L.149, L.278-279)

Response: The abbreviations have been corrected. See line 145-153. For the line 278-279, the abbreviations have been displayed in the manuscript.

(13) FLOWERING LOCUS C (DELLA)? -> FLOWERING LOCUS C (FLC)

Response: It is our mistake. It is FLC gene, and the sentence has been corrected. See line 52.

(14) Figure 11b gene-Jman007G0196700. MS3 is lacking in the X-axis.

Response: The MS3 has been added to the Figure 11b.

(15) L.144 J. mandshurica should be written in italic.

Response: It has been corrected. See line 145.

(16) L.759 and 761 TaKaRa, Beijing, China? Takara, Kyoto, Japan? Are they different company? It is strange for me.

Response: It has been corrected. See line 764.

(17) Figure 7. Phytohormone name can be shown as abbreviations.

Response: The Figure 7 has been corrected.

(18) The reviewer highly recommends that English grammar of the text is checked by native specialist before submitting to the journal.

Response: Thanks for your comments. The English grammar of the manuscript has been corrected by the Bullet Edits Limited with the number of 7bc4846856e46917de800eb0db08e41f.

Reviewer 3 Report

In the manuscript, the authors did characterization of phytohormones and transcriptomic profiling of the female and male flower development in Manchurian walnut (Juglans mandshurica Maxim.). In this study, phytohormones and transcriptomic sequencing analyses at the three stages of female and male flower growth were performed to understand the regulatory functions underlying flower development. Gibberellin is the most dominant phytohormone that regulates flower development. In total, 14,579 and 7,188 differentially expressed genes were identified after analyzing the development of male and female flowers, respectively, wherein 3,241 were commonly expressed. Enrichment analysis for significantly enriched pathways suggested the roles of MAPK signaling, plant hormone signal transduction, and sugar metabolism. Genes involved in floral organ transition and flowering were obtained and analyzed; these mainly belonged to the M-type MADS-box gene family. Three flowering-related genes (SOC1/AGL20, ANT, and SVP) strongly interacted with transcription factors in the co-expression network. Two essential CO genes (CO3 and CO1) were identified in the photoperiod pathway. Authors also identified two GA20xs genes, one SVP gene, and five AGL genes (AGL8, AGL9, AGL15, AGL19, and AGL42) that contributed to flower development. The findings are expected to provide a genetic basis for the studies on the regulatory networks and reproductive biology in flower development for J. mandshurica.

The manuscript is very well written and can be accepted after minor changes.

It would be nice if the author tried to validate at least one gene found in this study functionally.

I have found plagiarism in this article at L194-195, L198-200, L201-203, L251-252, L256-257, L261, L277-278, L300-301, L305, L310-311, L316-319, L325-326, L328-329, L394-396, L436, L442-443, L446-449, L452-453, L536-537, L545-546, L690-692, L693-694, L703, L711-712, L725-726. Please clean it.

Author Response

Thank you for your letter again and for the reviewers’ comments concerning our manuscript entitled “Characterization of phytohormones and transcriptomic profiling of the female and male flower development in Manchurian walnut (Juglans mandshurica Maxim.)” (ijms-1701820). Those comments are all valuable and very helpful for revising and improving the quality our paper. We have studied all comments carefully and have made corrections which we hope meet the standard of your highly esteemed journal. The main corrections in the paper and the responses to the reviewers’ comments are given below:

(1) It would be nice if the author tried to validate at least one gene found in this study functionally.

Response: Thanks for your comments. We would like to validate the key genes from Juglans mandshurica in Arabidopsis thaliana in the future.

(2) I have found plagiarism in this article at L194-195, L198-200, L201-203, L251-252, L256-257, L261, L277-278, L300-301, L305, L310-311, L316-319, L325-326, L328-329, L394-396, L436, L442-443, L446-449, L452-453, L536-537, L545-546, L690-692, L693-694, L703, L711-712, L725-726. Please clean it.

Response: Thanks for your comments. The contents above were mainly the GO/KEGG annotation information. Similar description could be found in previous study [1-3].

[1] Nefissi Ouertani, Rim, Dhivya Arasappan, Tracey A. Ruhlman, Mariem Ben Chikha, Ghassen Abid, Samiha Mejri, Abdelwahed Ghorbel, and Robert K. Jansen. 2022. Effects of Salt Stress on Transcriptional and Physiological Responses in Barley Leaves with Contrasting Salt Tolerance[J]. International Journal of Molecular Sciences 23, 9: 5006. https://doi.org/10.3390/ijms23095006

[2] Qiao, Qian, Chong Wu, Tian-Tian Cheng, Yu Yan, Lin Zhang, Ying-Lin Wan, Jia-Wei Wang, Qing-Zhong Liu, Zhen Feng, and Yan Liu. 2022. Comparative Analysis of the Metabolome and Transcriptome between the Green and Yellow-Green Regions of Variegated Leaves in a Mutant Variety of the Tree Species Pteroceltis tatarinowii[J]. International Journal of Molecular Sciences 23, 9: 4950. https://doi.org/10.3390/ijms23094950

[3] Pan, Zhao-Jun, Ya-Chi Nien, Yu-An Shih, Tsun-Ying Chen, Wen-Dar Lin, Wen-Hsi Kuo, Hao-Chun Hsu, Shih-Long Tu, Jen-Chih Chen, and Chun-Neng Wang. 2022. Transcriptomic Analysis Suggests Auxin Regulation in Dorsal-Ventral Petal Asymmetry of Wild Progenitor Sinningia speciosa[J]. International Journal of Molecular Sciences 23, 4: 2073. https://doi.org/10.3390/ijms23042073

Round 2

Reviewer 2 Report

The reviewer confirmed that the authors improved the manuscript according to the comments.